# Comparison of Microbial Diversity of Two Typical Volcanic Soils in Wudalianchi, China

**DOI:** 10.3390/microorganisms12040656

**Published:** 2024-03-26

**Authors:** Qingyang Huang, Fan Yang, Hongjie Cao, Jiahui Cheng, Mingyue Jiang, Maihe Li, Hongwei Ni, Lihong Xie

**Affiliations:** 1Institute of Natural Resources and Ecology, Heilongjiang Academy of Sciences, Harbin 150040, China; huangqingyang@163.com (Q.H.); yangfan81039@163.com (F.Y.); hjcao781228@163.com (H.C.); chengjiahui407@163.com (J.C.); 13804585147@163.com (M.J.); xielihong903@163.com (L.X.); 2Forest Dynamics, Swiss Federal Institute for Forest, Snow and Landscape Research, CH-8903 Birmensdorf, Switzerland; maihe.li@wsl.ch; 3Key Laboratory of Geographical Processes and Ecological Security in Changbai Mountains, Ministry of Education, School of Geographical Sciences, Northeast Normal University, Changchun 130024, China; 4School of Life Science, Hebei University, Baoding 071002, China; 5Heilongjiang Academy of Forestry, Harbin 150040, China

**Keywords:** bacteria, fungi, lava, soil microbial communities, WDLC

## Abstract

Volcanic lava is an excellent model of primary succession, in which basalt-associated microorganisms drive the cycling of different elements such as nitrogen, carbon, and other nutrients. Microbial communities in volcanic soils are of particular interest for study on the emergence and evolution of life within special and extreme conditions. The initial processes of colonization and subsequent rock weathering by microbial communities are still poorly understood. We analyzed the soil bacterial and fungal communities and diversities associated with lava (LBL) and kipuka (BK) sites in Wudalianchi using 16S and ITS rRNA Illumina Miseq sequencing techniques. The results showed that soil physical and chemical properties (pH, MC, TOC, TN, TP, AP, DOC, and DON) significantly differed between LBL and BK. The Shannon, Ace, and Pd indexes of fungi in the two sites showed a significant difference (*p* < 0.05). The dominant bacterial phyla forming communities at LBL and BK sites were Acidobacteria, Proteobacteria, Actinobacteria, and Basidiomycota, and their differences were driven by Gemmatimonadetes and Verrucomicrobia. The dominant fungal phyla of LBL and BK sites were Ascomycota, Zygomycota, and Rozellomcota, which differed significantly between the two sites. The microbial communities showed extremely significant differences (*p* < 0.05), with MC, pH, and nitrogen being the main influencing factors according to RDA/CCA and correlation analysis. Microbial functional prediction analysis across the two sites showed that the relative abundance of advantageous functional groups was significantly different (*p* < 0.05). The combined results drive us to conclude that the volcanic soil differences in the deposits appear to be the main factor shaping the microbial communities in Wudalianchi (WDLC) volcanic ecosystems.

## 1. Introduction

Soil microorganisms are the foundation of the Earth’s biosphere. Soil microbial communities are highly important to the ecosystem due to their important role in the occurrence and development of soil [1]. Volcanic soils represent a small portion of the world’s soils (<1%); however, they are highly productive and form unique ecosystems throughout the world [2]. Volcanic soils are derived from volcanic lava and volcanic ash, which are rich in trace elements and metal elements, and are entirely different from other soil types around the world, leading to different microbial activities [3]. Studies have shown that soil microbes commence colonizing the rock surface after lava congeals [4], contributing to early soil formation and initial community succession [5], and the entire volcanic soil formation process is closely related to soil carbon, nitrogen, and phosphorus content [6].

The biogeochemical cycle and vegetation regeneration succession change after a volcanic eruption. The primary succession starts from bare rock [7] and includes various ecological processes, such as those associated with vegetation, soil, and microorganisms. Studies found in the literature mainly relate to the characterization of volcanic soil formation, soil physicochemical properties, soil development and plant vegetation, and vegetation dynamics [8,9,10,11]. However, research investigating soil microbial populations is scarce. Previous studies have reported that the microbial communities in older volcanic soils are more structurally diverse than those in younger volcanic soils [12,13]. In the Fuji volcano and Wudalianhi volcanoes (WDLC), volcanic soil microbial biomass and diversity were both significantly low in the early stages of volcanic succession, which was affected by soil development time and soil nutrients [14,15]. Using high-throughput sequencing technology, the same results were found in the Wulanhada volcanic group, but the younger volcanic soils possessed a more complex microbial community network than older volcanic soils [16].

Changes in soil properties following volcanisms may have a substantial effect on soil microbes [17]. Zhang et al. found that the abundance, compositions, and functions of soil bacterial and fungal communities showed significant differences in the two volcanic habitats (the cone and crater). Particularly, the presence of soil physicochemical properties in the cone led to a more complex network and a larger number of keystone taxa [18]. Extreme environments can preserve some special microbiomes, potentially playing a critical role in soil fertility and ecosystem stability [19]. Wulanhada volcanic soils may have enriched some specific microorganisms closely related to the metabolism of soil carbon, nitrogen, and phosphorus, and then guided microbial multifunctionality [16]. The dominant bacteria, such as Proteobacteria, Acidobacteria, and Actinobacteria, in WDLC volcanoes are related to the igneous basalt from the core, indicating that the soil bacterial community structure is affected by the igneous rock matrix [20].

Volcanic eruption disturbances have crucial effects on terrestrial natural ecosystems through the generation and airborne transmission of volcanic ash, magma, gases and other ejecta produced [6]. As a member of the tentative list of World Natural Heritage in China, WDLC volcanic features are diverse and well-preserved, with their youthful morphology and scenic lava landforms. The two newest volcanoes, Laohei shan and Huoshao shan, erupted during 1719–1721 and formed new lunge lava landforms with an area 24,580 km^2^ [21]. There are also kipuka in the lava landforms in WDLC, with an island of land that survives volcanic eruptions when lava flows around it, but is not covered by later lava flows [22]. Both special types of soil exist in WDLC national park at the same time. Consequently, the present study is an attempt to investigate the soil properties of volcanic soils; in addition, the variation in microbial community structures were studied more closely in lava and kipuka ecosystems.

## 2. Materials and Methods

### 2.1. Site Description and Soil Sampling

The WDLC National Park (125°45′–126°30′ E; 48°30′–48°50′ N) covers an area of 988 km^2^ and is located at the northwest of the Heilongjiang Province of China (Figure 1). The mean annual temperature and precipitation were −0.5 °C and 476.33 mm, respectively. Volcanic ash soil comprised the main soil type in the WDLC.

A warm-temperate conifer-broadleaved-mixed forest was the original zonal vegetation type in the WDLC [7]. However, when Laohei shan and Huoshao shan erupted and destroyed the original vegetation and soil, the primary succession was re-established from bare rock. After 290 years, Ass *Larix gmelinii* and *Betula platyphylla* has become the prominent forest type in volcanic lava zones, while Ass *B. platyphylla* is the dominant forest type at the kipuka site (Figure 2).

In terms of the Larix-Betula-lava (LBL: 48°39′ N, 126°16′ E) and Betula-kipuka (BK: 48°44′ N, 126°06′ E) sites, due to there being no soil profile obtained for the LBL site, six sites were selected to collect surface (0–5 cm) soil samples from. The sampling was carried out in August 2020, and at each site, after litter removal, three samples of soil were collected and mixed together in order to obtain a representative composite sample to perform the chemical and microbiological analyses.

### 2.2. Chemical and Microbiological Analyses

Soil moisture content (MC) was determined by drying duplicate subsamples at 105 °C overnight. pH was measured using the potentiometric method (PB-10, Sartorius, Göttingen, Germany). Total organic carbon (TOC) and nitrogen (TON) were determined using an elemental analyzer (EA3000, Euro Vector, Foggia, Italia). The total phosphorus (TP) mass fraction of the soil was determined using molybdenum antimony colorimetry. The available phosphorus (AP) fraction of the soil was determined using NaHCO_3_ Extraction Colorimetry. Dissolved organic C (DOC) and N (DON) were determined using TOC analyzer (Multi N/C 2100S, Analytik Jena AG, Jena, Germany) [23,24].

DNA was extracted using the Power Soil DNA Isolation Kit following the protocol provided by the manufacturer (MoBio, Carlsbad, CA, USA). The V3–V4 regions of the 16S rDNA gene were amplified using barcoded primers 515F (5′-GTFYCAGCMGCCG CGGTAA-3′) and 806R (5′-GGACTACNVGGGTWTCTAAT-3′) for soil bacterial communities, and the ITS1 and ITS2 regions of the ITS rDNA gene were amplified using ITS1F (5′-CTTGGTCATTTAGAGGAAGTAA-3′) and ITS2R (5′-GCTGCGTTCTTCATCGATG C-3′) primers for soil fungal communities (GeneAmp^®^ 9700, ABI, New York, NY, USA). The PCR products were run on a 2% agarose gel, further purified using an AxyPrep DNA Gel Extraction Kit (Axygen Biosciences, Union City, CA, USA), and quantified using QuantiFluor™-ST (Promega, Madison, WI, USA) according to the manufacturer’s protocol. The purified amplicons were pooled equally and paired-end sequenced (2 × 300) on an Illumina MiSeq platform (Illumina, San Diego, CA, USA) according to standard protocols by Majorbio Bio-Pharm Technology Co., Ltd. (Shanghai, China). The raw sequencing data were compiled by the NCBI SRA database under the accession numbers PRJNA722363 and PRJNA722146.

### 2.3. Statistical Analysis

The Wilcoxon signed-rank test was used to test significant differences in the soil properties and microbial diversity index between LBL and BK. Differences were considered significant at *p* < 0.05, with marginal significance set at *p* < 0.01. All analyses were performed using SPSS 19.0 for Windows. Figures were generated using the Origin 8.0 package. Data are reported as the mean ± se.

We calculated the Sobs, Ace, Shannon, and Pd indices to assess species richness and diversity of soil bacterial and fungal communities based on the Operational Taxonomic Unit (OTUs) data [25]. Redundancy analysis (RDA) and canonical correspondence analysis (CCA) were also used to examine the relationship among the soil biochemical variables (pH, MC, TOC, TN, TP, AP, DOC, DON) and the soil microbial community compositions. We conducted forward selection of the environmental variables that were significantly correlated with variations in the microbial communities using the stepwise regression method and the significance test. The RDA and CCA were processed using R software (version 3.5.1), and we obtained two-dimensional ordination graphs.

## 3. Results

### 3.1. Soil Physico-Chemical Properties

Soil pH values ranged from 5.88 to 6.40 (Table 1). The soil pH value in BK was the most acidic with 5.88, while LBL had the highest soil pH value, with 6.40. The concentrations of MC, TOC, TN, DOC, and DON were significantly higher in BK than in LBL, while the concentrations of TP and AP were significantly higher in LBL than in BK.

### 3.2. Soil Microbial Diversity

In total, the bacterial data set encompassed 1,738,296 sequences. The average number of bacterial sequences was 96,572, with a mean length of 270 bp. For fungi, the data set encompassed a total of 1,296,852 sequences; the average number of fungal sequences was 72,047, with a mean length of 260 bp. The calculated sampling coverage showed no significant difference between LBL and BK (Table 2). The coverage values in the two soil types were over 0.996, indicating that the sequencing depth was adequate to cover most microorganisms and even included some rare species.

For bacteria, there were no differences in the alpha diversity indexes (Sobs index, Shannon index, Ace index, and Pd index) between LBL and BK sites (Table 2). For fungi, these indexes were significantly lower in LBL than in BK, whereas the Shannon index did not differ between the two sites.

### 3.3. Diversity and Community Composition

There were nine bacterial phyla with relative abundances above 1% (Figure 3). Proteobacteria, Actinobacteriota, and Acidobacteria were the dominant phyla, accounting for 72.91–79.81% of the total number of bacteria, followed by Verrucomicrobia, Planctomycetes, Bacteroidetes, Chloroflexi, and Gemmatimonadetes (Figure 3a). The abundances of Verrucomicrobia at the phylum level, Spartobacteria at the class level (Appendix A), and Chthoniobacterales at the order level (Appendix A) were significantly lower in LBL than those in BK (*p* < 0.01). Further, the difference in Verrucomicrobia at the phylum level resulted from differences in Chthoniobacterales at the order level and Spartbacteria at the class level. The abundance of Gemmatimonadetes at the phylum level was significantly higher in LBL than in BK (*p* < 0.01, Figure 3a). Their abundance differences between sites can be further identified at the order (Appendix A), family (Appendix A), and genus level (Appendix A).

In terms of the fungal soil communities, Ascomycota (51.36–79.08%) and Basidiomycota (17.81–25.66%) were the most dominant phyla in all samples (Figure 3b). At the phylum level, the abundance of Ascomycota, Zygomycota, and Rozellomcota differed significantly between LBL and BK sites (*p* < 0.05; Figure 3b); their differences can be further identified at the class (Appendix A), order (Appendix A), family (Appendix A), and genus level (Appendix A).

### 3.4. Correlations among Soil Nutrients, Microbial Community Composition

Pearson correlation analysis results showed that physicochemical properties significantly affected the dominant bacterial and fungal communities (*p* < 0.05). At the phylum level of bacterial species (Figure 4a), Verrucomicrobia and Nitrospirae were positively correlated with the MC, TOC, TN, DOC, and DON (*p* < 0.05), and Gemmatimonadetes was positively correlated with the pH, TP, and AP (*p* < 0.05). At the phylum level of fungi (Figure 4b), Rozellomycota and Zygomycota were positively correlated with the MC, TOC, TN, DOC, and DON (*p* < 0.01), and Ascomycota was positively correlated with the pH, TP, and AP (*p* < 0.05).

The data regarding the effects of soil nutrients on the major microbial community composition at the OTUs were analyzed using RDA/CCA analysis. The first two RDA axes explained 80.57% and 64.16% of the total variations in the major soil bacterial and fungal communities, respectively (Figure 5). Further, soil bacterial and fungal community structures between the LBL and BK treatments were clearly separated by the first principal components.

All soil factors had significant effects on the first two RDA axes of soil bacterial and fungal communities (*p* < 0.05). The degree of the influence of soil physicochemical factors on soil bacterial community structure was ranked as follows: MC > DON > pH > TN > TOC > AP > TP > DOC (Table 3). The degree of influence on fungal community structure was ranked as follows: MC > DON > pH > TOC > DOC > TN > TP > AP.

### 3.5. Function Prediction

FAPROTAX and FUNGuild were used to predict and annotate the function of soil microbial communities. There were 11 functional groups with relative abundance greater than 1% in both bacterial and fungal communities (Table 4). For the bacterial communities, chemoheterotrophic and aerobic chemoheterotrophic species comprised the major functional groups, and these were observed at a significantly higher abundance in LBL than in BK (*p* < 0.05). Functional groups involved in nitrogen cycling processes, such as nitrification, aerobic ammonia oxidation, nitrate respiration, and nitrogen respiration were also significantly higher in BK than in LBL (*p* < 0.05). For the fungal communities, undefined saprotrophs, ectomycorrhizal species, and plant-wood saprotrophs were the major fungal functional groups. Except for ericoid mycorrhizal, other fungal functional groups were significantly different between sites (*p* < 0.05).

## 4. Discussion

### 4.1. Soil Properties in LBL vs. BK

In WDLC, we found that soils at LBL sites with sparse vegetation cover had higher pH levels and lower soil moisture, compared with soils covered by denser vegetation in BK. Other analyses in the Tibetan plateau, French alpine and Antarctic regions, and along receding glacier chronosequences, all strongly supported the above results [26]. The soil water content in LBL is significantly lower than in BK, which is due to the shallow soil layer of the new volcanic plot and the fact that the permeable properties of basalt lava and volcanic ash reduce the surface runoff and cause insufficient soil moisture [7,27]. Meanwhile, the soil pH in LBL is significantly higher than in BK. The surface litter formed by the sparse broad-leaved dwarf forest in LBL is sparse [28], which may be the reason for the higher pH. The research results of Cui et al. [29] support this.

Many studies have indicated that soil properties are significantly influenced by some environmental factors such as soil type, vegetation type, and soil development time [30,31]. Soil in LBL sites is mainly developed from basalt volcanic rocks [27], which indicates that the time of soil development and soil formation is only around 300 years. The soil of BK is mainly developed from Cretaceous sedimentary and tertiary granite rocks, which can be occasionally found in uncovered volcanic lava [32]. In this study, as the time of soil development and soil weathering increased, the concentrations of DOC, TN, DOC, and DON increased; thus, these concentrations were significantly higher in BK than in LBL. These results are similar to what was reported by Fu et al. [33]. However, the concentrations of TP and AP were significantly higher in LBL than in BK. This confirms that alkali basalt and higher P concentrations are correlated with volcanic materials and the mineral substrate in WDLC volcanoes [33].

### 4.2. Diversity and Structures of Microbial Communities in LBL vs. BK

Previous studies have shown that volcanic eruptions can affect the structure and composition of soil microbial communities [6]. In our study, the community structures of bacterial and fungal communities are significantly different between LBL and BK soils. This result is supported by studies that have shown that soil microbial community structure varies among Cockell secondary forest successions after volcanic eruptions [12]. Our results support the theory proposed by Odum [34] which states that older soil microbial communities would be more structurally diverse, an observation that has been echoed in other studies of ecosystem development [35]. The poorer nutrition of soil in LBL means that the microorganism community at such sites is simple.

The most abundant phyla (Proteobacteria, Acidobacteria, Actinobacteria, Verrucomicrobia, and Chloroflexi) identified in our samples have already been associated with basalt-hosted microbial communities [36]. The phylum Proteobacteria has been confirmed to be involved in enhanced weathering of basalt and exhibits the ability to grow in poor environments, and it is considered to be an important organism type in carbon and nitrogen cycling in basalt environments [37]. The phylum Verrucomicrobia has been found to be associated with basalt lava environments in volcanoes, where they greatly contribute to basalt weathering [38,39]. The two sites contain Rhizobiales, Planctomycetales, Sphingomonadales, Rhodospirillales, and Burkholderiales (Appendix A), which have been associated with basalt communities in lava formed soil [39,40].

In fungi communities, the most abundant phyla Ascomycota (75.31–82.66%) and the most abundant family Pseudeurotiaceae (17.43–31.78%) were found in significantly higher abundance in LBL than in BK (Figure 2 and Appendix A). This may be explained by the fact that many different kinds of lichen are widely distributed in basalt lava in LBL [41]; lichens are symbiotic associates of fungi (usually an Ascomycete) [42]. The new volcanic lava platform is characterized by less water and good permeability, properties which are consistent with the characteristics of Basidiomycetes and Ascomycetes, species that prefer soil permeability [26].

### 4.3. Relationships between Soil Microbial and Soil Properties

Pearson correlation analysis and RDA were applied to reveal the effect of soil properties on the dominant microbial community (Figure 4 and Figure 5). In this study, MC, DON, and pH were the main factors driving the variation of soil microbial community structure, and were strongly correlated with the abundance of Verrucomicrobia, Gemmatimonadetes, Ascomycota, and Rozellomycota. The abundance levels of these four phyla were significantly different between LBL and BK sites. These differences are attributed to the diversity in the soil properties and vegetation characteristics among microbial communities [43].

Soil moisture was the most important factor that controlled fungal community composition, as it indirectly changes microbial metabolic activity by affecting nutrient cycling and plant growth, and lower soil water content affects microbial metabolic activity [44]. However, pH can also change the soil environment by affecting the utilization rate of nitrogen, thus affecting the microbial community structure [45]. It is suggested that Verrucomicrobia might be related to nitrogen cycling in soil [46].

### 4.4. Potential Functional Groups of Soil Microbial

The nitrogen cycling functional groups of the soil in the WDLC volcanic habitat are mainly nitrogen fixation, nitrification, and aerobic ammonia oxidation. Nitrogen fixation was found to be dominated by *Bradyrhizobium* (Appendix A), in agreement with earlier studies [47]. *Bradyrhizobium* is an important nitrogen fixation symbiont, which can directly improve the availability of soil nutrients, soil productivity, and diversity of vegetation, and increase the abundance of soil microbial communities with specific functions [48]. In our study, the two functional groups involved in nitrification and aerobic ammonia oxidation mainly belong to *Nitrosospira* at genus level and Nitrosomonadaceae at family level (Appendix A). These are the main nitrifying functional bacteria, and can oxidize ammonia to NO_2_ [49]. Wang et al. reported that when soil moisture is not limited to soil oxygen transport, the digestion rate of nitrifying bacteria increases with an increase in soil moisture [50]. This can explain why the abundance of nitrifying bacteria in BK was significantly higher than that in LBL; oxygen content was higher in soil with higher moisture content, and this soil type was beneficial for the growth of good ammonia oxidizing bacteria.

Saprophytic fungi are important decomposers in soil and play an important role in nutrient cycling [51]. In this study, except the unclassified fungi, the dominant nutrient type of soil fungi in WDLC was saprophytic (26.16%), which is consistent with most previous studies [52,53]. Our research showed that the abundance of soil pathotrophic fungi in BK was significantly higher than in LBL, as the more suitable temperature and humidity environment in BK promoted the growth of pathogens [54]. Meanwhile, compared to coniferous trees, broad-leaved plants such as *Betula platyphylla* have higher levels of polyphenols in their leaves, which is closely related to the increase in the abundance of pathogens at BK sites [55].

## 5. Conclusions

There were significant differences in soil physicochemical properties, in the diversity, composition, and structure of soil microbial communities, and the microbial functional groups in LBL and BK sites that had undergone volcanic disturbances. The volcanic soil differences in the deposits appeared to be the main factor shaping the microbial communities in WDLC volcanic ecosystems. In addition to soil environmental factors, factors such as volcanic materials, litter composition, and vegetation type were also important factors driving soil microbial community construction. Therefore, the aim of our future research is to study the coupling relationships between vegetation, microorganisms, and soil environment, and further carry out multi-dimensional comprehensive experiments to explain the response of soil microbial community structure and function to environmental heterogeneity in volcanic ecosystems.

## Figures and Tables

**Figure 1 microorganisms-12-00656-f001:**
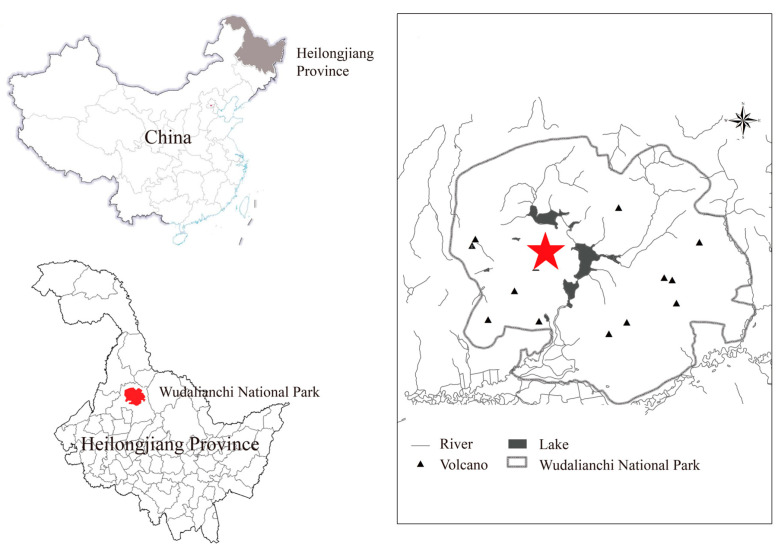
Overview of the WDLC Research Area. The asterisk indicates the study site in the WDLC National Park, Heilongjiang Province, China.

**Figure 2 microorganisms-12-00656-f002:**
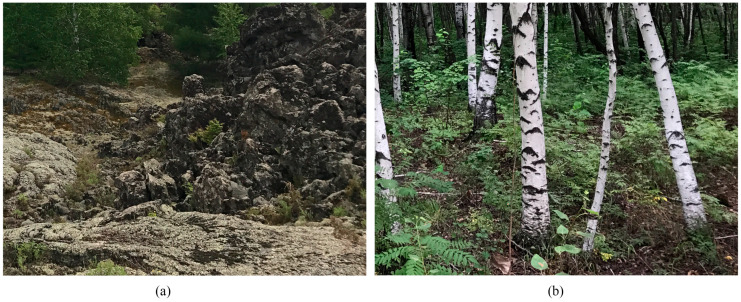
Study sites: (**a**) Larix-Betula-lava (LBL), Ass *Betula platyphylla* and *Larix gmelinii* in volcanic lava; (**b**) Betula-kipuka (BK), Ass *B. platyphylla* in kipuka site.

**Figure 3 microorganisms-12-00656-f003:**
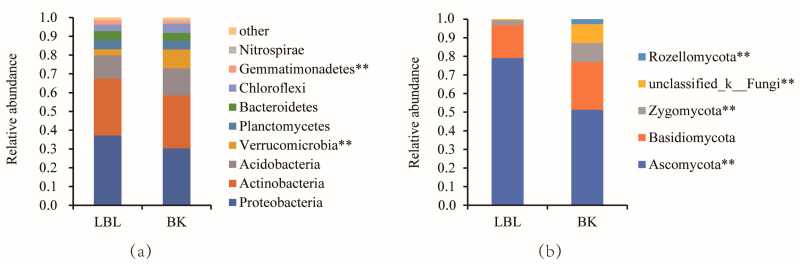
Bacterial (**a**) and fungal (**b**) community composition at the phylum. The phyla with a relative abundance of <0.1% in soils were presented in other phyla. (** *p* < 0.01) indicate significant differences between abundances in LBL and BK soils based on Wilcoxon signed-rank test.

**Figure 4 microorganisms-12-00656-f004:**
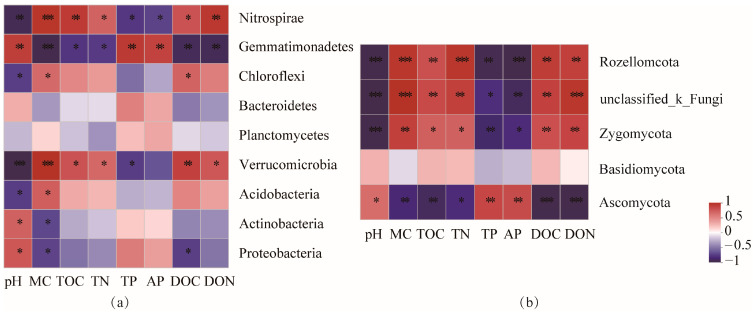
Pearson correlation coefficients describing the relationships between soil nutrients and acterial (**a**) and fungal (**b**) community composition. (*** *p* < 0.001), (** *p* < 0.01), and (* *p* < 0.05).

**Figure 5 microorganisms-12-00656-f005:**
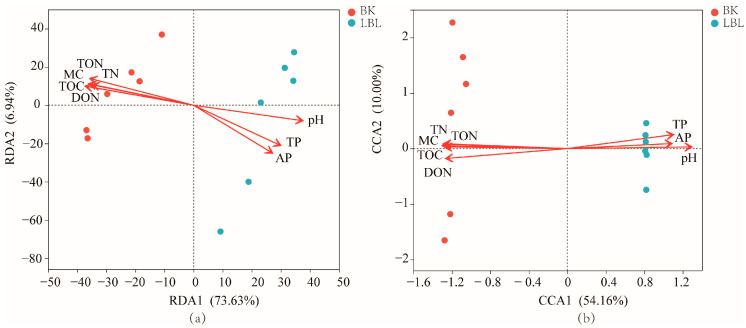
The RDA/CCA of soil bacterial community (**a**) and fungal community (**b**) based on phyla level with soil factors. Ordination plot showing the influence of the environmental variables (red line, red arrows) on the sample sites (circles). BK soils, red circles; LBL soil, blue circles.

**Table 1 microorganisms-12-00656-t001:** Soil physical and chemical characteristics in volcanic lava. Abbreviations: MC (moisture content), TOC (total organic carbon), TN (total nitrogen), TP (total phosphorous), AP (available phosphorous), DOC (Dissolved organic carbon), DON (Dissolved organic nitrogen). The data in the table represents mean ± standard error (*n* = 6). If letters are the same, the divergence is not significant; on the contrary, divergence is significant at 0.05 levels.

Sample	pH	MC (%)	TOC (g·kg^−1^)	TN (g·kg^−1^)	TP (g·kg^−1^)	AP (mg·kg^−1^)	DOC (g·kg^−1^)	DON (g·kg^−1^)
LBL	6.40 ± 0.01 ^a^	16.69 ± 0.94 ^b^	36.62 ± 1.40 ^b^	4.69 ± 0.31 ^b^	2.24 ± 0.03 ^a^	17.32 ± 0.45 ^a^	3.40 ± 0.02 ^b^	0.37 ± 0.01 ^b^
BK	5.88 ± 0.01 ^b^	53.12 ± 1.63 ^a^	212.43 ± 1.30 ^a^	16.90 ± 0.07 ^a^	1.68 ± 0.03 ^b^	11.23 ± 0.25 ^b^	6.96 ± 0.19 ^a^	2.72 ± 0.03 ^a^

**Table 2 microorganisms-12-00656-t002:** Diversity index of each soil sample at a cutoff level of 0.97. The data represents mean ± standard error (*n* = 6). If alphabet is same, the divergence is not significant; on the contrary, divergence is significant at 0.05 levels.

Sample/Index	Bacterial		Fungal	
	LBL	BK	LBL	BK
Sobs	2319.33 ± 43.94 ^a^	2360.83 ± 36.41 ^a^	205.00 ± 9.31 ^b^	281.00 ± 10.05 ^a^
Shannon	6.35 ± 0.11 ^a^	6.44 ± 0.02 ^a^	3.39 ± 0.14 ^a^	3.21 ± 0.34 ^a^
Ace	2413.91 ± 23.60 ^a^	2478.56 ± 32.03 ^a^	218.92 ± 9.05 ^b^	294.04 ± 11.51 ^a^
Pd	111.65 ± 1.82 ^a^	108.53 ± 2.26 ^a^	42.10 ± 2.22 ^b^	65.45 ± 2.88 ^a^
Coverage	0.9967 ± 0.0004 ^a^	0.9962 ± 0.0005 ^a^	0.9995 ± 0.0001 ^a^	0.9994 ± 0.0002 ^a^

**Table 3 microorganisms-12-00656-t003:** Significance tests between soil bacterial/fungal communities and soil physicochemical properties (* *p* < 0.05).

Soil Factors	Bacterial		Fungal	
	R^2^	*p*-Value	R^2^	*p*-Value
pH	0.8558	0.001 *	0.9840	0.004 *
MC	0.8773	0.002 *	0.9951	0.006 *
TOC	0.8056	0.002 *	0.9746	0.006 *
TN	0.8548	0.001 *	0.9544	0.004 *
TP	0.7823	0.002 *	0.7482	0.008 *
AP	0.7986	0.002 *	0.6946	0.012 *
DOC	0.7310	0.002 *	0.9568	0.003 *
DON	0.8623	0.002 *	0.9864	0.006 *

**Table 4 microorganisms-12-00656-t004:** Differences in potential functional groups with average relative abundance > 1% across the two sites. The data in the table represents mean ± standard error (*n* = 6). If alphabet is same, the divergence is not significant; on the contrary, divergence is significant at 0.05 levels.

Types	Functional Groups	LBL-Mean (%)	BK-Mean (%)
Bacterial	Chemoheterotrophy	34.32 ± 0.72 ^a^	28.67 ± 1.28 ^b^
	Aerobic chemoheterotrophy	33.92 ± 0.72 ^a^	28.28 ± 1.27 ^b^
	Nitrogen fixation	7.65 ± 1.09 ^a^	6.1 ± 0.5 ^a^
	Nitrification	1.39 ± 0.41 ^b^	4.12 ± 0.58 ^a^
	Aerobic ammonia oxidation	1.12 ± 0.3 ^b^	3.03 ± 0.37 ^a^
	Nitrate reduction	2.11 ± 0.19 ^a^	1.96 ± 0.14 ^a^
	Animal parasites or symbionts	0.63 ± 0.21 ^b^	3.13 ± 1.41 ^a^
	Ureolysis	1.93 ± 0.24 ^a^	1.43 ± 0.2 ^b^
	Aromatic compound degradation	1.11 ± 0.22 ^b^	1.85 ± 0.32 ^a^
	Nitrate respiration	1.19 ± 0.18 ^b^	1.42 ± 0.17 ^a^
	Nitrogen respiration	1.19 ± 0.18 ^b^	1.42 ± 0.17 ^a^
Fungal	Undefined Saprotroph	20.67 ± 5.3 ^a^	6.18 ± 1.5 ^b^
	Ectomycorrhizal	18.87 ± 4.5 ^a^	7.82 ± 2.24 ^b^
	Plant Saprotroph-Wood Saprotroph	25.47 ± 4.42 ^a^	0.01 ± 0.01 ^b^
	Endophyte-Litter Saprotroph-Soil SaproTroph-Undefined Saprotroph	2.54 ± 0.55 ^b^	15.96 ± 3.9 ^a^
	Endophyte	9.46 ± 2.52 ^a^	0.57 ± 0.09 ^b^
	Ericoid Mycorrhizal	4.45 ± 1.49 ^a^	5 ± 0.86 ^a^
	Ectomycorrhizal-Orchid Mycorrhizal- Root Associated Biotroph	1.19 ± 0.22 ^b^	4.77 ± 1.43 ^a^
	Animal Pathogen	0.14 ± 0.06 ^b^	5.18 ± 0.75 ^a^
	Animal Pathogen-Undefined Saprotroph	1.07 ± 0.19 ^b^	3.9 ± 0.91 ^a^
	Fungal Parasite-Undefined Saprotroph	1.26 ± 0.24 ^a^	0.66 ± 0.19 ^b^

## Data Availability

Data are contained within the article and Appendix A.

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
