# Peer review of "Comparison of Microbial Diversity of Two Typical Volcanic Soils in Wudalianchi, China"

_microorganisms, 2024, doi:10.3390/microorganisms12040656_

Round 1
Reviewer 1 Report
Comments and Suggestions for Authors
The article is interesting but needs several corrections as follows:
- the material and methods section needs improvements, please add references to all methods or give details about the instruments used for physical measurements
- please describe better the results about the bacterial communities. Please review the order of Tables and also the citation of supplementary figures in manuscript. There is no Fig S5! but I suppose is about S4 in the text.
- please give explanation on figures from supplementary file.
- please add explanation of all abbreviations when used first.
- please review the last part of conclusion.
-please check carefully the English language.

The English language should be also reviewed.
Author Response
We appreciate your kind suggestions in manuscript accordingly. We have carefully revised the manuscript according to your suggestions. Please see the attachment.

Reviewer 2 Report
Comments and Suggestions for Authors
Dear Editor and Authors
The manuscript brings an interesting approach, is well written and presents appropriate material and methods.
In general, I consider that few adjustments should be revised before publication in Microorganisms.
Please include the geographic coordinates of the sample points.
It is required to present in which public database the results were deposited.
Author Response
Thank you very much for your recognition of our manuscript. We have revised the manuscript according to your suggestions. Please see the attachment.
